# Therapeutic Effect of a Latent Form of Cortistatin in Experimental Inflammatory and Fibrotic Disorders

**DOI:** 10.3390/pharmaceutics14122785

**Published:** 2022-12-13

**Authors:** Jenny Campos-Salinas, Margarita Barriga, Mario Delgado

**Affiliations:** Institute of Parasitology and Biomedicine López-Neyra, IPBLN-CSIC, Parque Tecnológico de la Salud, 18016 Granada, Spain

**Keywords:** cortistatin, inflammation, neuropeptides, prodrug, latency-associated peptide, sepsis, inflammatory bowel disease, pulmonary fibrosis, scleroderma

## Abstract

Cortistatin is a cyclic neuropeptide that recently emerged as an attractive therapeutic factor for treating inflammatory, autoimmune, fibrotic, and pain disorders. Despite of its efficiency and apparent safety in experimental preclinical models, its short half-life in body fluids and its potential pleiotropic effects, due to its promiscuity for several receptors expressed in various cells and tissues, represent two major drawbacks for the clinical translation of cortistatin-based therapies. Therefore, the design of new strategies focused on increasing the stability, bioavailability, and target specificity of cortistatin are lately demanded by the industry. Here, we generated by molecular engineering a new cortistatin-based prodrug formulation that includes, beside the bioactive cortistatin, a molecular-shield provided by the latency-associated protein of the transforming growth factor-β1 and a cleavage site specifically recognized by metalloproteinases, which are abundant in inflammatory/fibrotic foci. Using different models of sepsis, inflammatory bowel disease, scleroderma, and pulmonary fibrosis, we demonstrated that this latent form of cortistatin was a highly effective protection against these severe disorders. Noteworthy, from a therapeutic point of view, is that latent cortistatin seems to require significantly lower doses and fewer administrations than naive cortistatin to reach the same efficacy. Finally, the metalloproteinase-cleavage site was essential for the latent molecule to exert its therapeutic action. In summary, latent cortistatin emerges as a promising innovative therapeutic tool for treating chronic diseases of different etiologies with difficult clinical solutions and as a starting point for a rational development of prodrugs based on the use of bioactive peptides.

## 1. Introduction

Cortistatin (CST) is a cyclic peptide discovered in 1996 in the rat cerebral cortex and has the ability to inhibit cortical activity [1]. Although it is mainly produced by GABAergic interneurons of the cerebral cortex and hippocampus [2], cortistatin is also expressed by some cells of peripheral tissues, including endothelial cells, endocrine cells, peripheral nociceptive neurons, smooth muscle cells, and a variety of immune cells (B and T lymphocytes, monocytes, macrophages, and dendritic cells) [3,4,5,6]. Encoded by the *Cort* gene, cortistatin is secreted in two forms of 17 and 29 amino acids (CST-17 and CST-29) in humans, or of 14 and 29 amino acids (CST-14 and CST-29) in mice and rats [3]. Cortistatin shows high structural and sequence homologies with somatostatin. Thus, CST-14 shares 11 of its 14 residues with somatostatin-14, including the two Cys-Cys residues required for their cyclic conformation and the amino acid sequence (FWKT) that interacts with all the somatostatin receptors Sst1–Sst5 [7]. Consequently, cortistatin shares with somatostatin some pharmacological and functional properties, including decrease of cAMP production and inhibition of multiple hormone release, cell proliferation, neuronal activity, angiogenesis, and pain [3,7]. However, cortistatin, but not somatostatin, shows the capacity to bind to a variety of G protein-coupled receptors, other than Sst1-5, such as the growth hormone secretagogue receptor (GHSR1, also known as ghrelin receptor), the human Mas gene-related receptor X-2 (MrgX2) and truncated forms of the Sst5 [8,9,10,11], and probably a still unidentified cortistatin-specific receptor. These alternative signaling pathways confer to cortistatin many functional properties that are not shared by somatostatin or the Sst-analogues used in clinic, including the induction of slow wave sleep, reduction of locomotor activity, inhibition of vascular calcification and smooth muscle cell function, and regulation of many key events of innate and adaptive immune responses [5,7]. From a therapeutic point of view, the fine tuning exerted by cortistatin in the immune system has lately emerged as one of the most attractive functions of this neuropeptide. Thus, a growing number of reports have demonstrated the protective effect of cortistatin in several experimental preclinical models of inflammatory and organ-specific autoimmune disorders, including sepsis, lung inflammation, inflammatory bowel disease (IBD), multiple sclerosis, rheumatoid arthritis, uveitis, and myocarditis [5,12,13,14,15,16,17,18]. These therapeutic actions were exerted by inhibiting a plethora of inflammatory cytokines and cytotoxic factors, by impairing the production of chemokines and the recruitment of neutrophils, monocytes, and lymphocytes, and by inducing immune tolerance through the restoration of the balance between autoreactive and regulatory T lymphocytes [5,12]. Moreover, recent evidence showed that cortistatin exerts potent anti-fibrotic activities in a variety of tissues and organs and protects against chronic fibrotic disorders of diverse etiologies, independently of its anti-inflammatory activity [15,19,20].

Although cortistatin has proved to be an effective and safe therapy in many preclinical models of disease [5,12,13,14,15,16,17,18,19,20], and even in the treatment of human disorders [21,22], it faces with two drawbacks that could impair its translation into clinical practice in a medium-time period. First, as occurs with other neuropeptides, due to its rapid degradation by endopeptidases, cortistatin has a short half-life in body fluids and tissues (approximately 2 min in plasma) [23], which implies repetitive administrations of high doses of the peptide to reach optimal effects in vivo. Second, with the promiscuity of cortistatin for several binding sites that are expressed in a variety of cells in multiple locations in the body, the risk of generating pleiotropic and undesired side effects after its injection arises. In order to increase the half-life of the peptide, several strategies have been previously proposed and assayed for other immunomodulatory peptides, like vasoactive intestinal peptide (VIP) or melanocyte-stimulating hormone (MSH), including residue substitutions and cyclic-structures in the molecule for improving its stability, encapsulation in nanoparticles or micelles, or the use of viral vectors for increasing its protection [24,25,26,27,28]. In this sense, we recently generated a stable analogue of cortistatin by selective amino acid substitutions of the naïve peptide that showed more than ten-fold higher half-life in plasma and similar therapeutic effects compared to cortistatin in experimental chronic IBD [23]. While these strategies significantly solve the problem of reduced stability, they do not avoid the potential emergence of undesired side effects. This issue has received attention lately after recent findings suggesting that increased activity of the cortistatin-MrgX2 system could be associated with itch and chronic prurigo in humans [29,30]. Therefore, the design of a cortistatin-based compound that acts as a prodrug, which protects cortistatin from peptidase-induced degradation and specifically releases the bioactive peptide at the target site, is desirable. In this sense, Adams and collaborators in 2003 designed by molecular engineering a latent form of the cytokine interferon-β [31] that was later adapted to VIP and MSH [32] to demonstrate in vivo their increased efficiency for the treatment of experimental inflammatory disorders. Following this strategy, in this study, we have designed a latent form of cortistatin that consists of a molecular shield provided by the latency-associated peptide (LAP) of the transforming growth factor-β1 (TGF-β1) linked to the sequence of cortistatin by a cleavage site of recognition by metalloproteinases (MMP) (Figure 1a), in which LAP protects cortistatin from degradation by tissue endopeptidases once injected and avoids its binding to receptors and the MMP-cleavage site by allowing the release of the bioactive cortistatin only in places in the body where these enzymes are abundant, like foci of inflammation or fibrosis (Figure 1b). The efficiency of this new latent form of cortistatin with potential increased bioavailability was assayed in various experimental models of inflammatory and fibrotic disorders.

## 2. Materials and Methods

### 2.1. The Animals and Ethic Statement

C57Bl/6 and Balb/c mice were purchased from Charles River (Barcelona, Spain). The experiments reported in this study followed the ethical guidelines for investigations of experimental animals approved by the Animal Care and Use Board and the Ethical Committee of Spanish Council of Scientific Research (Number 1101/2021) and were performed in accordance with the guidelines from Directive 2020/569/EU of the European Parliament on the protection of animals used for scientific purposes. Animal studies are reported in compliance with the ARRIVE guidelines [34]. All animals were housed in a controlled-temperature/humidity environment (22 ± 1 °C, 60–70% relative humidity) in individual cages (10 mice per cage, with wood shaving bedding and nesting material), with a 12 h light/dark cycle (lights on at 7:00 AM) and fed with rodent chow (Global Diet 2018, Harlan) and tap water ad libitum. According to the protocol established by the experimental animal unit of the IPBLN-CSIC, mice were allowed to acclimatize to housing room for a minimum of 5 days after their reception in the animal facility and then from the housing room to the experimental room for 1 h before performing each experiment. Mice were randomly assigned to the different experimental groups (the number of animals per group and the names of each experimental group are indicated below in the corresponding methodological section or in the figure legends). None of the animals were excluded from the study. The Ethical Committee established as humanitarian end points the observation of a sustained body weight loss higher than 20% for two days (especially important for mice subjected to sepsis, colitis, and pulmonary fibrosis models), impossibility of the animal to access food and water (even facilitated in the cage bed), evident signs of pain (assessed by maintained audible groans), and/or signs of limb mutilations. When indicated, animals were sacrificed by carbon dioxide affixation (in all cases, death was ensured by further exsanguination).

### 2.2. Reagents

Unless otherwise indicated, all reagents used in this study were purchased from Sigma-Aldrich (St. Louis, MO, USA). Bleomycin sulfate (with specific activity of 1.6–2.0 U/mg) was dissolved in saline solution (0.9% NaCl) at 1 mg/mL (1.8 U/mL) and stored at −20 °C and was diluted in saline at the indicated doses immediately before its injection in animals. Mouse cortistatin-29 (from Bachem, Bubendorf, Switzerland) was dissolved in ddH_2_O and stored at −80 °C at a dose of 0.1 mM and was diluted in ddH_2_O (used as vehicle) at the indicated dose and volume immediately before its use in the experimental models.

### 2.3. Cloning and Expression of Latent-CST

To generate a latent form of cortistatin, we proceeded as follows. First, we amplified the sequence comprising the signal peptide (SP, 1-29aa) and LAP of TGFβ1 prepropeptide (30-274aa, NM_000660.7) from the plasmid SPLAPTGFB1-pCMV6-XL4 (OriGene Technologies GmbH, Herford, Germany) by PCR using specific primers (Forward: AAGCTTATGCCGCCCTCCGGGCTGCGG and Reverse: GAATTCGCTTTGCAGATGCTGGGCCCTCTCCA) containing restriction sites for *Hind*III and *Eco*RI. The amplified SP-LAP fragment was cloned in the plasmid pcDNA3.1 + (Thermo Fisher) in order to generate a parenteral plasmid, further referenced in this manuscript as pLAP, which was used as backbone to subsequently clone the other elements to synthesize the latent form of cortistatin (Figure 1a) using the In-fusion cloning technique (Takara Bio, Saint-Germain-en-Laye, France) following the manufacturer’s recommendations. Thus, a double-stranded DNA molecule was synthesized (Mtabion, Planegg, Germany), including the nucleotide sequence for the first flexible linker L1 (aa-sequence: GGGGS), the cleavage site that is recognized by MMPs (aa-sequence: PLGLWA), the second flexible linker L2 (aa-sequence: GGGGSAAA), and cortistatin-29 (NM_012835.1), flanked at both ends by two 16-bp fragments that included the sequence of recombination with the pLAP and recognition sites for the *Eco*RI restriction enzyme (see Table 1). In all cases, the reading frame of these primers was adjusted to remain in phase and to preserve the *Eco*RI restriction site. This plasmid will be further referenced in this manuscript as pLatent-CST. In addition, we generated two control plasmids that excluded the MMP-cleavage site, a pLAP-CST that did not include the sequences for L1, MMP-site recognition and L2, and a pLAP-L1L2-CST that included L1 and L2 sequences, but the MMP-site recognition was deleted (Figure 1b, see Table 1 for sequence).

Plasmids coding for LAP, Latent-CST, and LAP-CST were transfected into human embryonic kidney 293 cells (HEK293, Sigma-Aldrich) using the LipoD293 DNA in vitro transfection reagent (SignaGen Laboratories, Frederick, MD, USA), following the manufacturer’s recommendations. Cells were cultured in complete DMEM medium (DMEM supplemented with 10% fetal bovine serum, 100 U/mL penicillin/streptomycin, 2 mM L-glutamine, 4.5 g/mL glucose, all from Gibco) in 75 cm^2^ Nunc Flask (Nunc, Thermo Fisher, Waltham, MA, USA) at 37 °C and 5% CO_2_. Culture supernatants were collected 24 h and 48 h after cell transfection, assayed for the contents of cortistatin-29 using a specific competitive ELISA for cortistatin (Phoenix Pharmaceuticals, Karlsruhe, Germany) and stored at −80 °C until their use for treating animal models. Whereas supernatants of pLAP-transfected cells showed undetectable amounts of cortistatin-29, cells transfected with pLatent-CST, pLAP-L1L2-CST, or pLAP-CST produced 8–11 ng/mL of cortistatin-29 at 24–48 h of culture (Figure 2a). For treating the different experimental models described below, we used pooled supernatants from different pLatent-CST-, pLAP-L1L2-CST- or pLAP-CST-transfected cell cultures (collected at 48 h), containing 10 ng/mL (approximately 3 pmol/mL) of cortistatin-29 in the latent form. As a control of reference, we used the same volume of culture supernatants of pLAP-transfected HEK293 cells. HEK293 cells are one of the few human cell lines approved by the European Medicines Agency and US Food and Drug Agency for the manufacture of recombinant protein therapeutics, and beside this regulatory approval, HEK293 cells offer multiple advantages that convert them as very attractive for the pharmaceutical companies as expression systems for recombinant protein production [35]. In our case, HEK293 cells allow the posttranslational modifications and proteolytic processing of the LAP-associated prepropeptides and their secretion in mature dimeric forms. This strategy was successfully used by other authors to produce latent forms for other immunomodulatory peptides, including MSH, VIP, and interferon-β [31,32,36].

To evaluate the stability of cortistatin in the different forms, supernatants collected at 48 h from cultures of HEK293 cells that were transfected with pLatent-CST, pLAP-L1L2-CST, or pLAP-CST were incubated for 24 h, 48 h, 72 h, and 120 h in 20% bovine serum-supplemented complete DMEM medium at 37 °C, and the content of cortistatin was determined by ELISA at the indicated time points (Figure 2b).

To evaluate the effect of MMP in the content of cortistatin in the latent form, supernatants (1 mL) of pLatent-CST-transfected HEK293 cells (collected at 48 h culture) were incubated in complete DMEM medium in the absence or presence of MMP-1 (300 ng/mL, Sigma-Aldrich) at 37 °C, in continuous shaking (at 150 rpm) for 12 h. After incubation, the medium was centrifuged (at 2500× *g* for 2 h at 4 °C) through Amicon ultra-4 centrifugal filter units (with Ultracell-10 membrane, Millipore). The contents of cortistatin at the initiation and at the end of culture in the retained fraction in the filter (containing the latent form, >10 kDa) and the eluted fraction were determined by ELISA.

### 2.4. Induction and Treatment of Sepsis

Polymicrobial-induced sepsis was induced by the surgical procedure of cecal ligation and puncture (CLP) as previously described [37]. Briefly, female C57BL/6 mice (22–24 g body weight, 9–12 weeks-old) were anesthetized (i.p., ketamine 80 mg per kg mouse, Richter Pharma; xylazine 10 mg per kg mouse, Fatro Iberica) and a small abdominal midline incision was made. The cecum was exposed, ligated approximately 8 mm from the cecal tip with suture (6-0 silk) and punctured through both surfaces twice with a 22-gauge needle. The stool was extruded (1 mm) throughout the punctures, the cecum placed back into its normal intra-abdominal position and the abdomen and skin were closed by layers with a running suture (6-0 silk). Animals in which cecum was exposed but not subjected to CLP were used as sham controls. All animals received subcutaneous resuscitative normal saline (20 mL/kg body weight) 4 h after surgery. Animals were treated 6 h and 18 h after CLP with i.p. injections of 400 µL of culture supernatants of cells transfected with either pLAP (without cortistatin-29) or pLatent-CST (containing 1.2 pmol of cortistatin-29 in latent form). Animals treated i.p. with 400 µL of PBS (vehicle) or cortistatin-29 (0.5 nmol) at 6 h and 18 h post-CLP were used as controls of reference. In addition, to investigate the involvement of MMP-cleavage site, animals with CLP were treated i.p. with 400 µL of supernatants of cells transfected with pLAP-CST or pLAP-L1L2-CST (containing 1.2 pmol of cortistatin-29 in latent form, without MMP-cleavage site). Survival was monitored daily for 10 days. Clinical signs were scored at day 3 after CLP in a blinded manner using a scale from 0 to 12 that evaluate in each animal six signals (each individually scored from 0 to 2) including: piloerection, colitis, reduced movement, trembling, rheum in the eyes, crouched position. In an independent set of experiments, animals were sacrificed at 48 h after CLP, serum was obtained by cardiac puncture, lungs were dissected, and peritoneal fluid was collected by peritoneal lavage with cold PBS-EDTA. Interleukin-6 (IL-6) and monocyte chemoattractant protein-1 (MCP-1or CCL-2) contents in cell-free peritoneal lavage fluids and sera were quantified by specific sandwich ELISAs using capture/biotinylated detection antibody obtained from BD Biosciences (San Jose, CA, USA) and Peprotech (London, UK) according to the manufacturer’s recommendations. The amount of nitric oxide (NO) in peritoneal fluid was estimated from the accumulation of nitrite, a stable NO metabolite, by the Griess assay as previously described in detail [38]. For histopathological evaluation, freshly collected lungs were fixed in 10% buffered formalin, embedded in paraffin and sectioned. Cross-sections (5-μm) were stained with hematoxylin/eosin (H&E) as previously described [15,23]. Images were acquired in an Axio Scope.A1 microscope (Carl Zeiss, Germany) using 5X and 10X objectives and 10X ocular and analyzed with Zen 2011 Light Edition software (Carl Zeiss) in a blinded manner by at least two independent researchers in whole-lung sections [15]. CLP-induced histopathology was scored determining the extent of inflammatory cell infiltration on alveolar walls, alveolar hemorrhage, and alveolar septum congestion, using a semiquantitative scale from 0 (normal and no focal inflammatory infiltrates) to 4 (severe infiltration and damage in lung structure).

### 2.5. Induction and Treatment of IBD

Chronic colitis was experimentally induced in slightly anesthetized (induced with 4% isoflurane) male Balb/c mice (7–8 weeks old, 25–27 g body weight) by repetitive intrarectal (i.r.) administrations of 100 µL of the haptenizing reagent 2,4,6-Trinitrobenzene sulfonic acid (TNBS, 2.5 mg per mouse, dissolved in 50% ethanol) every seven days, for three weeks, as previously described [23]. Animals intrarectally injected with 100 µL of 50% ethanol, instead of TNBS, were used as basal controls of reference. For treatment, animals were i.p. injected with 400 µL of PBS (untreated, vehicle) or with 400 µL of culture supernatants of cells transfected with either pLAP (without cortistatin) or pLatent-CST (containing 1.2 pmol of cortistatin-29 in latent form) at 24 h after each TNBS infusion. Animals were daily monitored for the appearance of diarrhea, body weight loss, and survival. At different time points, colitis signs were scored in base of stool consistency and rectal bleeding by two blinded observers using the following scale: 0, normal stool appearance; 1, slight decrease in stool consistency; 2, moderate decrease in stool consistency; 3, moderate decrease in stool consistency and presence of blood in stools; 4, severe watery diarrhea and moderate/severe bleeding in stools.

### 2.6. Induction and Treatment of Experimental Scleroderma

Skin fibrosis was induced in female C57BL/6 mice (6–8 weeks-old) under slight anesthesia (induced with 4% isoflurane) by repetitive intradermal (i.d.) injections of bleomycin (3.2 U per kg mouse, in 100 µL of saline, three times per week, for four weeks) in a localized place in the right side of saved back of mice as previously described [19]. Moreover, saline (in 100 µL volume) was repetitively injected i.d. in a localized point in the left side of the mouse back, contralateral to bleomycin-induced lesion (used as basal control of reference). Treatment consisted in subcutaneous (s.c.) injections of 100 µL of culture supernatants of cells transfected with either pLAP (without cortistatin) or pLatent-CST (containing 0.3 pmol of cortistatin-29 in latent form), every seven days, for four weeks, starting five days after the first bleomycin injection, around the bleomycin-induced skin lesion. We used as reference controls of treatment, animals that were s.c. injected with 100 µL of PBS (untreated, vehicle) or cortistatin-29 (1 nmol), three times per week, for four weeks, starting five days after the first bleomycin injection, around the bleomycin-induced skin lesion. Four weeks after the first bleomycin injection, animals were sacrificed and skin (0.7 cm^2^ of the bleomycin-induced lesion and of the saline-injected contralateral site) and lungs were dissected, fixed in 10% buffered formalin, embedded in paraffin and sectioned. Skin and lung cross-sections (5 µm) were stained with Masson’s trichrome using standard technique and images were acquired in an Axio Scope A1 microscope using 5X objective and 10X ocular and analyzed with Zen 2011 Light Edition software in a blinded manner by two independent researchers/pathologists as previously described [19]. Skin fibrosis was determined in at least three sections per biopsy by measuring dermal thickness (distance in µm from the base membrane of epidermal layer to the hypodermal junction with subcutaneous fat, determined as the mean of three random measurements in each section) using the Fiji-ImageJ software (http://imagej.net/Fiji, accessed on 30 May 2017). Lung fibrosis was scored in whole lung sections according to a semi-quantitative scale (0 to 4) that evaluates alveolar thickness, damage of lung structure, and fibrosis extension as previously described [19]: 0, normal lung or minimal fibrous thickening of alveolar or bronchial walls; 1, moderate thickening of the wall, with less than 25% of fibrotic area, but without obvious damage to lung architecture; 2, formation of fibrous bands, fibrous masses in 25–50% of lung area, and definitive damage of lung structure; 3, severe distortion of the structure and large fibrous areas (>50% of the cross-section involved); 4, total fibrous obliteration of the field. Results show the mean value of at least three no overlapping randomly selected areas per lung section (with 5X objective) and three representative sections per mouse (discarding at least 200 μm between sections).

### 2.7. Induction and Treatment of Experimental Lung Fibrosis

Severe lung fibrosis was induced in anesthetized (with ketamine and xylazine as described above) female C57BL/6 mice (6–8 weeks-old) by injecting intratracheally (i.t.) bleomycin (3.2 U per kg mouse, in 50 μL of saline) as previously described [19]. Treatment consisted in the intranasal (i.n.) infusion onto the nares of 10 μL of culture supernatants of cells transfected with either pLAP (without cortistatin), pLatent-CST (containing 0.03 pmol of cortistatin-29 in latent form) or pLAP-CST or pLAP-L1L2-CST (containing 0.03 pmol of cortistatin-29 in latent form, but without MMP-cleavage site), once per week, for three weeks, starting five days after bleomycin injection. Survival was monitored daily. Lungs were dissected immediately after death of animals as a consequence of severe pulmonary fibrosis or they were taken from sacrificed mice at the end of experiment and processed for paraffin sectioning. Histopathological signs were scored in at least three sections per lung as described above. In addition, the presence of activated myofibroblasts in fibrotic lungs was determined by immunofluorescence analysis of the expression of α-smooth muscle actin (α-SMA) as previously described [19]. In brief, Formalin-fixed lung sections were incubated in 10-nM sodium citrate/0.05% Tween 20 (20 min, 100 °C) for antigen retrieval, cooled in water, and then incubated twice during 5 min in PBS/0.025% Triton X-100. Sections were blocked with 10% goat serum/1% BSA (120 min, 20 °C) and incubated with primary mouse anti-mouse αSMA antibody (Sigma-Aldrich, clone 1A4, diluted at 1:1000 in PBS/1% BSA, overnight, 4 °C). After extensive washing with PBS/0.025% Triton X-100, sections were incubated with secondary Alexa Fluor 568-goat anti-mouse antibody (Life Biotechnologies/Thermo Fisher, diluted at 1:1000 in PBS/1% BSA, 60 min, 20 °C). Nuclei were DAPI counterstained (Sigma-Aldrich, diluted at 1:1000 in PBS, 5 min, 20 °C) and sections were mounted in Mowiol. Sections in which we omitted primary antibody were used as negative controls, showing in all cases a lack of fluorescence signal. Sections were examined in an Olympus IX81 fluorescence microscope (Olympus Life Science, Hamburg, Germany) and the images were acquired at 100× magnification (Olympus CellSens Imaging v1.12 software) using the same parameters and region of interest (ROI) between samples and were quantified for the mean of fluorescence intensity (integrated density in grey scale) using the Fiji ImageJ software (five random areas per section, two sections per mouse).

### 2.8. Data and Statistical Analysis

All experiments are randomized and blinded. All data are expressed as mean ± SD of mice/experiment, with the exception of the time curves in the IBD model that are expressed as mean ± SEM in order to increase the clarity in the graphs. We analyzed data for statistical differences between two groups (i.e., cytokine levels) by the unpaired two-tailed Student’s *t*-test or the non-parametric Mann–Whitney U-test (for data that do not reach normal distribution). Normality distribution of data was assessed using the D’Agostino and Pearson omnibus test. For multiple group comparisons, we used regular one-way ANOVA and post-hoc Bonferroni’s test for dermal thickness, or the non-parametric equivalence Kruskal–Wallis analysis of variance test and post-hoc Dunn´s test for sepsis clinical score, fibrosis score, and histopathological score. For multiple comparisons of time curves, we used repeated measure ANOVA and post-hoc Bonferroni’s test for body weight loss and non-parametric repeated measure ANOVA (Friedman test) for colitis score. Survival curves were analyzed with the Kaplan–Meier log-rank test. All analyses were performed using GraphPad Prism v5.0 software (La Jolla, CA, USA). Differences were considered statistically significant if the alpha probability was 0.05 or less.

## 3. Results

### 3.1. Treatment of Experimental Sepsis and IBD with Latent Cortistatin

We initially evaluated the therapeutic potential of the newly generated latent form of cortistatin in experimental polymicrobial peritonitis, which is widely considered as a reliable preclinical model of human sepsis and in which cortistatin was previously reported to exert a therapeutic effect [16,39]. In untreated mice, ligation and puncture of the cecum caused a mortality rate around 75% (Figure 3b) due to disseminated intravascular coagulation with multiple organ failure, as indicated by severe congestion, hemorrhage, fibrin deposits, edema, thrombosis, and massive accumulation of leukocytes in lungs (Figure 2c) and other target organs (not shown). This pathology is a consequence of the inflammatory response against intestinal bacteria that systemically disseminated once they colonized the peritoneal cavity. As expected, i.p. treatment with Latent-CST, at 6 h and 18 h hours after CLP induction (see scheme in Figure 3a), attenuated clinical manifestations of sepsis, such as lethargy, diarrhea, body weight loss, and hypothermia (Figure 3d) and significantly improved survival (Figure 3b) by reducing the histopathological signs that are characteristics of this disorder (Figure 3c). In contrast, injection of LAP failed to show any therapeutic effects in septic animals (Figure 3b,c). As previously reported [16,23], treatment with naïve cortistatin showed similar protective actions in CLP-induced sepsis, although at a dose 400-fold higher than that observed for the latent form (Figure 3b,c). Similar to naïve cortistatin [39], treatment with Latent-CST reduced the production of a panel of inflammatory and oxidative mediators at the local and systemic levels (Figure 3e). Interestingly, two single injections of the prodrug were enough to induce long-lasting protective effects.

We further investigated the therapeutic potential of Latent-CST in an established experimental model of chronic IBD induced by i.r. administration of TNBS in 50% ethanol, which displays clinical, histopathological, and immunological features of human Crohn’s disease [40], and in which we previously described curative effects for cortistatin and its stable analogues [23,41]. In this colitis model, intestinal inflammation results from the initial ethanol-induced rupture of the intestinal barrier that is followed by the TNBS-mediated haptenization of autologous host mucosal proteins and subsequent stimulation of a Th1 cell-mediated immune response against TNBS-modified self-antigens. Mice subjected to three consecutive cycles of intracolonic TNBS infusion (Figure 4a) developed a severe and chronic illness characterized by a sustained body weight loss (Figure 4b) that is accompanied by significant bloody diarrhea, rectal prolapse, and extensive pancolitis (Figure 4c), resulting in a mortality of 80% (Figure 4d). A single systemic injection of Latent-CST at the beginning of each cycle of TNBS (Figure 4a) significantly reduced the body weight loss, improved the colitis signs, and doubled the survival rate (Figure 4b–d). As expected, the administration of empty LAP did not show beneficial effects in TNBS-induced colitis (Figure 4).

Therefore, a therapy that is based in the use of the latent form of cortistatin emerges as a potential complementary tool for treating human sepsis and IBD.

### 3.2. Protective Effect of Latent-CST in Experimental Fibrosis

Recent evidence has identified cortistatin as an endogenous anti-fibrotic factor in many tissues, including lung, skin, and liver, and that it shows potent therapeutic actions in a variety of chronic fibrotic disorders that affect these tissues [15,19,20]. The anti-fibrotic effects of cortistatin are directly exerted by regulating the activation fibroblasts and their differentiation into myofibroblasts, the major cell effectors in pathologic fibrosis, independently of its anti-inflammatory activity [15,20]. Therefore, we next investigated the capacity of the latent form of cortistatin to mitigate the exacerbated fibrosis that occurs in response to tissue injury. We firstly used an experimental model of skin fibrosis that is induced by repetitive intradermal injections of the antineoplastic drug bleomycin and mirrors many histopathological signs found in human systemic sclerosis [42]. Systemic sclerosis, also known as scleroderma, is a human autoimmune disorder that affects connective tissue, leads to fibrosis of the skin and internal organs (lungs, heart, kidneys, and gastrointestinal tract), and often results in death [43]. In this experimental model [42], the injury of epithelial and endothelial cells by bleomycin induces the release of profibrotic cytokines and growth factors that in turn activate dermal fibroblasts to secrete excessive extracellular matrix proteins, favor the infiltration of inflammatory cells, and partially deplete the dermal vascular system (Figure 5b). The perilesional injection of naïve cortistatin or of Latent-CST, but not of LAP, notably improved bleomycin-induced skin fibrosis, measured by a significant reduction of dermal thickness, extracellular matrix deposition, and inflammatory infiltration (Figure 5b). It is important to highlight that the latent form of cortistatin reached an efficacy similar to naïve cortistatin at significantly lower doses (0.3 pmol Latent-CST versus 1 nmol cortistatin) and required fewer administrations (once per week versus three times per week). As previously described for cortistatin [19], the fact that we delayed the initiation of Latent-CST treatment to a time point in which inflammatory infiltration already occurred, supports an action independent from its potential effect in the immune response that accompanies the progression of skin lesion.

Furthermore, pulmonary fibrosis is one of the extradermal manifestations of scleroderma, being suffered by around 40% of patients and it is a major cause of morbidity and mortality [43]. The scleroderma model used in this study may also co-occur with lung fibrosis [19,42]. In fact, we observed that lungs of mice that were intradermally challenged with bleomycin had evident signs of fibrosis with moderate thickening of the alveolar walls and damage of tissue architecture and around 30% of fibrotic area (Figure 5c). Subcutaneous injection of cortistatin or Latent-CST, but not of LAP, significantly reduced these histopathological signs of fibrosis in the lungs (Figure 5c). These findings point to the therapeutic potential of the latent form of cortistatin in chronic fibrotic diseases.

### 3.3. The MMP-Cleavage Site Is Essential for Latent-CST to Be Active in Ameliorating Inflammation and Fibrosis

Finally, to evaluate whether the release of cortistatin from the latent form was needed for exerting its therapeutic effect, we generated two peptides (LAP-CST and LAP-L1L2-CST) that lack the recognition site by MMP while maintaining the other functional elements of the latent molecule (Table 1, Figure 1). As expected, in contrast to that observed for Latent-CST, the treatment with LAP-CST failed to exert any protective effect on experimental sepsis (Figure 6a). Moreover, in a severe form of pulmonary fibrosis induced by intratracheal injection of bleomycin that caused a 100% mortality, the injection of LAP-CST or LAP-L1L2-CST did not improve survival rate (Figure 6b) or the histopathological score and fibrotic markers in lung parenchyma, which showed marked thickening of the alveolar septa, almost devoid of alveolar space, extensive areas of pulmonary parenchyma obliteration (with more than 60% of lung occupied by dense fibrosis) (Figure 6c), and presence of αSMA-expressing myofibroblasts in peribronchiolar dense fibrotic areas (Figure 6d). However, the same amount of Latent-CST significantly reduced the mortality caused by severe pulmonary fibrosis (Figure 6b–d), as previously described for high-doses of naïve cortistatin [22]. These findings indicate that the insertion of the MMP-cleavage site in Latent-CST is essential for exerting its protective effect in chronic inflammation and fibrosis. Finally, to investigate whether cortistatin could be cleaved from the latent form by MMP, we incubated Latent-CST in serum-supplemented medium in the absence or presence of recombinant MMP-1, and after 12 h of incubation the medium was filtered through a size-exclusion filter and both the retained (containing Latent-CST) and eluted fractions were assayed for cortistatin-specific ELISA. Whereas we detected stable levels of cortistatin in Latent-CST incubated in a serum-supplemented medium for a long period of time (Figure 1b), the addition of MMP-1 during 12 h drastically reduced more than 70% of the cortistatin content in the retained fraction (cortistatin levels: 8.29 ng/mL in the absence of MMP-1 versus 2.61 ng/mL in the presence of MMP-1, performed in duplicate). Although we detected low amounts of cortistatin in the eluted fraction, probably as a consequence of the degradation of cortistatin during the long incubation period, we found four times higher levels of cortistatin in the MMP-1-treated samples (0.16 ng/mL vs. 0.04 ng/mL in the presence or absence of MMP-1, respectively). These results indirectly suggest that cortistatin was released from the latent form by MMPs.

## 4. Discussion

Although evidence in preclinical and clinical studies has supported the efficiency and safety of cortistatin-based treatments in humans and animals [5,12,13,14,15,16,17,18,19,20,21,22,23], the pharmaceutical industry is still demanding new tools and strategies that increase the bioavailability and specificity of cortistatin for its use in the treatment of a variety of chronic disorders. In this study, we designed, by molecular engineering, a latent form of cortistatin that tries to overcome the two main drawbacks of treatments based on naïve cortistatin: its short half-life and its potential pleiotropic effects. Our data indicate that the new formulation of cortistatin is as effective as naïve cortistatin in protecting against acute and chronic disorders caused by severe inflammation and fibrosis. These therapeutic effects are apparently observed at doses between 400 (in systemic pathway) and 3000 (in local pathway) times lower and with significantly fewer administrations than those required for the naïve peptide, which supposes an evident advantage from a therapeutic point of view, mainly in a chronic scenario. Because the content of bioactive cortistatin in the latent form was quantified by ELISA, and LAP could act as a shield interfering with the recognition of cortistatin epitopes by antibodies, we could be underestimating the real content of cortistatin in the supernatants. Since the cortistatin-specific ELISA is based on the use of a polyclonal serum, it is probable that some of the epitopes of the cortistatin molecule are exposed in the LAP shell and recognized by the antibodies. Moreover, we detected similar cortistatin levels in the three different latent forms expressing the bioactive peptide designed in this study, showing homogeneity in the determination of cortistatin independently of the presence of other functional elements, such as flexible linkers or MMP-cleavage site. In this sense, Vessillier et al. [32] also used competitive ELISAs to measure the content of MSH and VIP in culture supernatants of HEK293 cells that were transfected with plasmids expressing these anti-inflammatory peptides in LAP-based latent forms, reaching similar results to those obtained in our study with cortistatin. In fact, the injection of purified MSH in latent form showed significantly increased therapeutic efficiency in an experimental model of peritonitis, at doses >30-fold lower than those observed for the naïve peptide [32]. In any case, this is a potential limitation of this study, and in the absence of other types of measurements that confirm the concentrations of cortistatin secreted by transfected HEK293 cells with all serum components and the exact content of the peptide in the injected suspension, we should be cautious establishing comparative rates of efficiency between naïve cortistatin and its latent form. Moreover, the effects of both peptides were only directly compared based on a single dose in some of the experimental models, and, therefore, relative potency should be solely established after a further analysis of a face-to-face in vivo dose-response in the same study. Furthermore, we have directly injected culture supernatants collected from transfected HEK293 cells for treating animals, instead of using, as others [31,32,36], purified latent forms or gene-based therapies by intramuscular injection of the plasmid and in vivo electroporation. Although the use of culture supernatants from genetically modified cells is widely used as a therapeutic strategy in many experimental models, and although HEK293 cells are considered a safe and clean cellular system for producing recombinant therapeutic proteins [35] and we have included in all our experiments appropriate negative controls (i.e., culture supernatants from HEK293 cells that were transfected with empty LAP), further preclinical studies using purified Latent-CST from serum-free transfected HEK293 cell cultures will be necessary to confirm our results, before they can be translated into clinical practice in animal or human health. 

Our study also demonstrated that the latent form of cortistatin is only effective when an MMP-cleavage site is included in the molecule. These findings could indicate that the cleavage of the bioactive cortistatin from the whole molecule by the MMPs that are present in the inflammatory and fibrotic foci is a requisite for its binding to specific receptors expressed in immune and fibrotic cell mediators and for exerting a therapeutic action. The protection of cortistatin by the molecular shield provided by LAP would probably avoid an optimal interaction of cortistatin with its specific receptors (mainly Sst2, Sst4, and GHSR1) and the subsequent blockade of proinflammatory and profibrotic signaling (mainly NF-kB and Smad2/3, respectively). This possibility could not only explain the increased efficiency that was observed for latent cortistatin in comparison to naïve cortistatin, but also indicates an improvement in its safety after a systemic administration of the compound. This means that naïve cortistatin needs to be injected at high doses to rapidly reach the target cell at enough concentrations for signaling before being degraded, and it needs to be repeatedly administrated to maintain a long-term response. Cortistatin in the latent form, however, is protected from degradation until it is released at the inflammatory/fibrotic foci, easily reaching a concentration close to the effective dose (around 1–10 nM) previously reported to regulate immune and fibrotic cells and to the *Kd* (around 0.5–5 nM) of the receptors involved in the response of cortistatin in these cells [5,12,16,44]. In this sense, LAP would have a dual function, shielding cortistatin from degradation and from its binding to receptors in undesired places. Although we indirectly demonstrated that the presence of MMP-1 seems to release cortistatin from the latent molecule, we did not investigate whether the released peptide is able to signal through Sst or GHSR in immune/fibrotic cells in vitro, and this is another limitation of our study that needs to be further addressed. Moreover, although we found stable concentrations of cortistatin in the latent form in serum-supplemented medium for a long period of time, the half-life and tissue distribution of the molecule after its in vivo administration remains unknown. In this sense, an in vivo half-life for the LAP-latent forms of VIP and interferon-β in the range of 30–50 h has been previously described, which is a 1200-fold increase compared to the reported few minutes of the corresponding free peptides [31,32].

Previous reports clearly demonstrated that the effect of cortistatin in IBD is mediated by downregulating the Th1-driven inflammation in the colonic mucosa and through the promotion of regulatory T cell-induced tolerance responses against self-antigens [41]. We could extrapolate these multilayer mechanisms found for cortistatin to the action of Latent-CST in this pathology, although this was not addressed in this study. In this sense, we confirmed that Latent-CST injection regulated many of the inflammatory and oxidative factors that naïve cortistatin previously decreased in experimental sepsis [16,17,39]. From a therapeutic point of view, it is important to mention that Latent-CST showed similar protective actions in chronic colitis to those previously reported for a stable cortistatin analogue, which displayed therapeutic efficiencies that are comparable to those observed for treatments of reference in the clinical management of patients with Crohn’s disease, such as anti-TNFα antibodies or mesalazine [23]. Although both peptides were not compared side-by-side in the same study, is is noteworthy that the stable cortistatin-like analogue reached the protective effect with significantly higher doses and more frequent injections [23] than those observed for Latent-CST (three injections per week of 6 nmol of cortistatin analogue versus one injection per week of 1.2 pmol of Latent-CST). Therefore, the use of Latent-CST opens up new possibilities for the treatment of patients with inflammatory bowel syndromes that fail to respond to other therapies.

Furthermore, it is not only important considering therapies based in the use of Latent-CST for treating dermal fibrosis in patients with scleroderma but also for avoiding the frequent appearance of serious pulmonary complications in these patients. Moreover, bleomycin is a chemotherapeutic drug widely used for treating many types of cancer, and because a significant number of bleomycin-treated patients develop severe side-effects that are associated with pulmonary and skin fibrosis, the systemic injection of the latent form of cortistatin could help to keep the adherence to the antineoplastic treatment in these patients.

Since we have shown the therapeutic activity of the latent form of cortistatin in a variety of inflammatory and fibrotic diseases, this promising therapy can be translated into many other chronic disorders where MMPs are involved, including to the control of inflammatory and neuropathic pain, in which cortistatin was reported to act as a potent multimodal analgesic factor [45,46,47] and with an especial interest of the industry in developing new Sst agonists [48]. Moreover, the strategy described in this study opens the possibility to increase even more the specificity of the release of the bioactive peptide with the insertion in the molecule of cleavage recognition sequences for MMP subtypes that are specifically present or more abundant in some tissues (i.e., the central nervous system, synovial tissue, lung) or disorders (i.e., multiple sclerosis, rheumatoid arthritis, pulmonary fibrosis) [49,50,51]. Finally, although the therapeutic effect of cortistatin in many disorders is exerted through its binding to more than a single receptor, mainly GHSR1 and Sst2/4, this being an advantage compared to somatostatin, in the case that signaling through a selective receptor is a preferable option, cortistatin could be substituted by the selected Sst- or GHSR1-specific peptidic agonist in the latent recombinant molecule.

In summary, the new cortistatin formulation that we characterize in this study could increase the chances of further bedside translation of cortistatin-based therapies and opens a path to the rational design of prodrugs to help control chronic inflammatory and fibrotic diseases of different etiologies that are characterized by elevated morbidity and mortality.

## 5. Patents

The data showed in this study are partially protected by patent: WO/2020/165457A1: Cortistatin or an analog, therefore, as a therapeutically active agent in latent form. Inventors: Mario Delgado and Jenny Campos-Salinas. Applicant: Consejo Superior de Investigaciones Cientificas (CSIC).

## Figures and Tables

**Figure 1 pharmaceutics-14-02785-f001:**
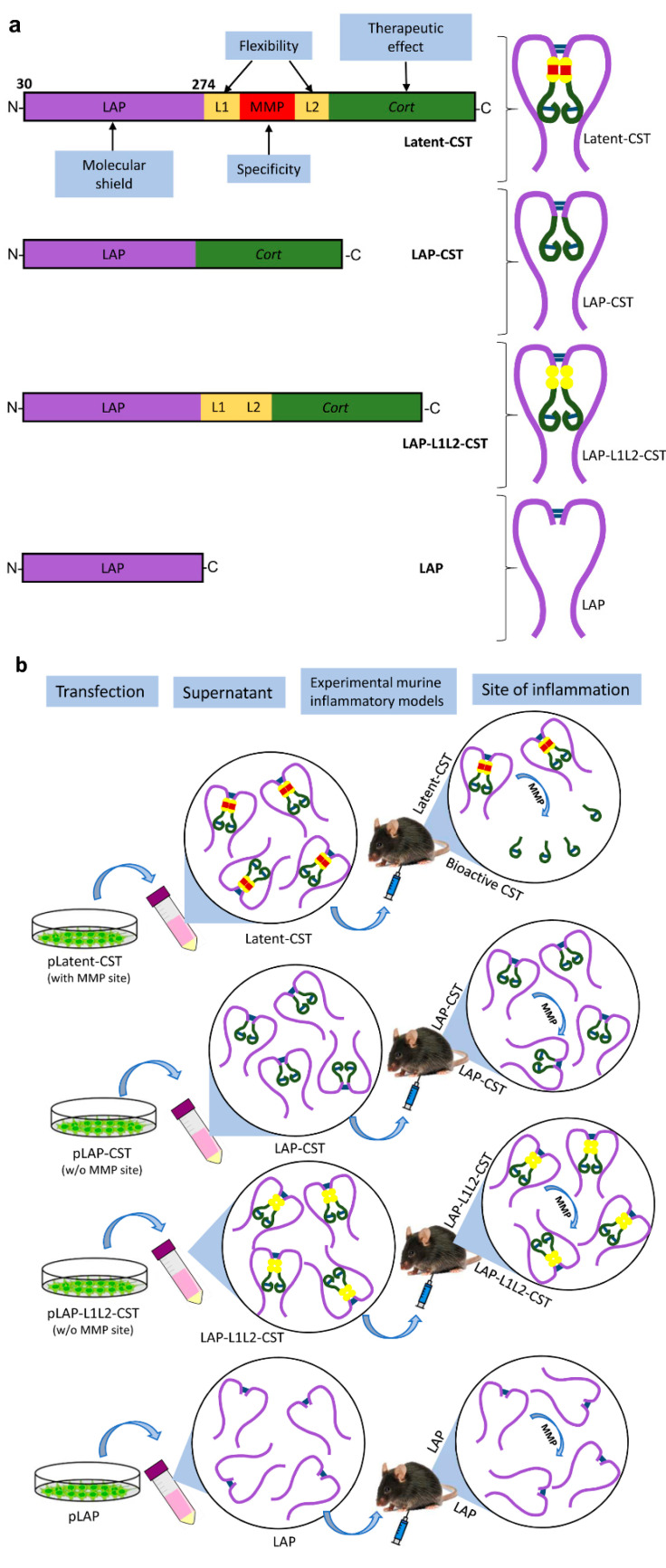
Experimental strategy used in this study for designing and evaluating the effect of a latent form of cortistatin. (**a**) Schematic representation of primary structure (left) and putative folding (right) of the different molecules used in this study, highlighting the different functional elements that are included in each peptide. pLatent-CST consists of the Latency-Associated Protein (LAP) of TGF-β1 followed by a first flexible linker L1 (aa-sequence: GGGGS), a cleavage site that is recognized by metalloproteinases MMPs (aa-sequence: PLGLWA), a second flexible linker L2 (aa-sequence: GGGGSAAA), and finally the sequence for murine cortistatin-29 (CST). LAP is not represented in scale in comparison to the other functional elements of the molecule (the number of amino acids of LAP is depicted, excluding a signal peptide of 29 residues). Flexible linkers are necessary for acquiring a suitable three-dimensional structure of the recombinant molecule. The MMP-specific recognition site allows the release of bioactive cortistatin from the latent form only at the inflammatory/fibrotic foci. pLAP-CST and p-LAP-L1L2-CST lack the MMP-specific cleavage site and are unable to release bioactive cortistatin. Empty pLAP was used as a control of reference. The molecule is depicted as a homodimer, assuming that LAP should dimerize through two disulfide bonds in endoplasmic reticulum during its secretory pathway, as previously described [33]. (**b**) Putative protein products obtained by the cleavage action of the MMPs at the inflammatory or fibrotic foci after their injection in the mouse.

**Figure 2 pharmaceutics-14-02785-f002:**
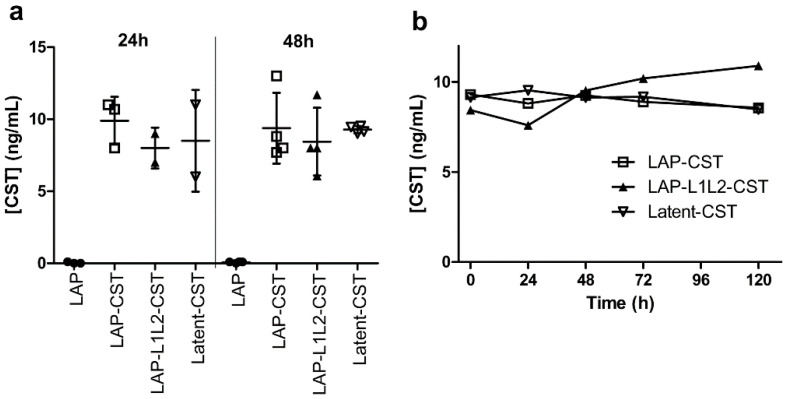
Content of cortistatin in the different peptide formulations. (**a**) Concentration of cortistatin in the culture supernatants of HEK293 cells transfected with pLAP, pLAP-CST, pLAP-L1L2-CST, or pLatent-CST after 24 h and 48 h of culture. Results are the mean±SD with dots representing individual values of biological independent cultures. (**b**) Concentrations of cortistatin in culture supernatants collected from HEK293 cells transfected with pLAP, pLAP-CST, pLAP-L1L2-CST, or pLatent-CST after 48 h of culture and additionally incubated for the indicated times (24 h, 48 h, 72 h, and 120 h) in serum-supplemented medium. Results represent the mean (n = 2–3 cultures, in duplicates).

**Figure 3 pharmaceutics-14-02785-f003:**
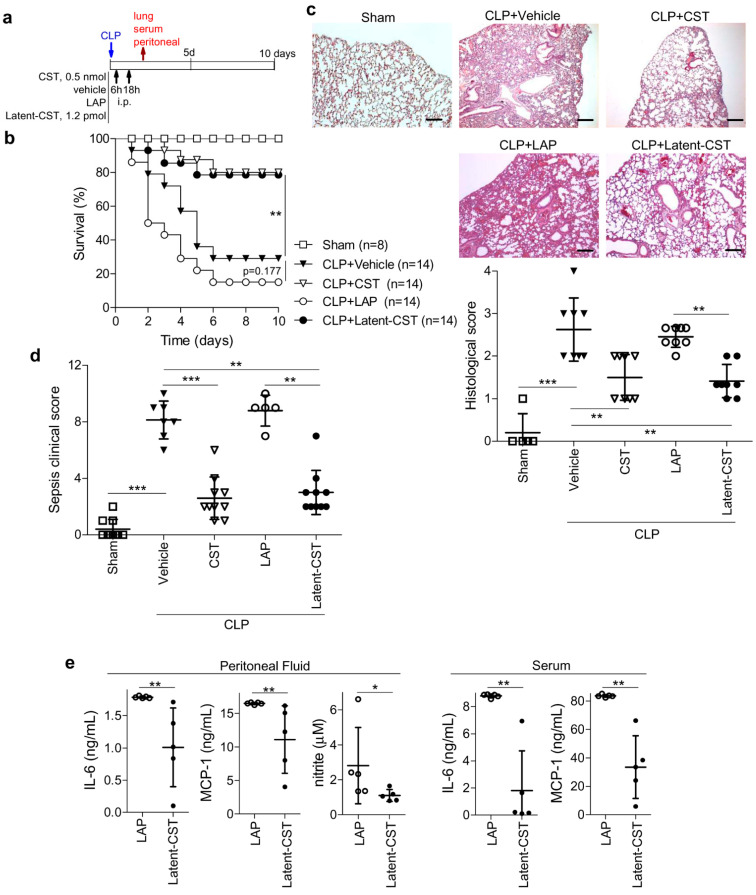
Latent cortistatin showed similar protection against sepsis to naïve cortistatin. (**a**) Mice with cecal ligation and puncture (CLP)-induced sepsis were treated as described in the scheme. Sham-operated animals were used as basal controls of reference. (**b**) Survival was monitored daily for 10 days (n = 8–14 mice/group, Kaplan–Meier log-rank test, ** *p* < 0.01 versus vehicle-treated septic mice). (**c**) The lung lobules were collected 48 h after CLP and histopathological signs were scored in hematoxylin/eosin-stained lung sections. n = 8 mice/group, one-way ANOVA Kruskal–Wallis test (*p* < 0.001) and post-hoc Dunn’s multiple comparisons, ** *p* < 0.01, *** *p* < 0.001. Representative images are shown (scale bar: 200 µm). (**d**) The clinical signs of sepsis were scored at day 3 by evaluating animal movement and posture, piloerection, trembling, and colitis. n = 5–10 mice/group, one-way ANOVA Kruskal–Wallis test (*p* < 0.001) and post-hoc Dunn’s multiple comparisons, ** *p* < 0.01 and *** *p* < 0.001. (**e**) The levels of inflammatory cytokines, chemokines, and nitric oxide (NO) were determined in the peritoneal fluid and serum collected 48 h after CLP surgery. n = 5 mice/group, unpaired two-tailed Student’s *t*-test or non-parametric Mann–Whitney U-test; * *p* < 0.05, ** *p* < 0.01 versus LAP-treated mice. Results are the mean ± SD with dots representing individual values of biologically independent animals.

**Figure 4 pharmaceutics-14-02785-f004:**
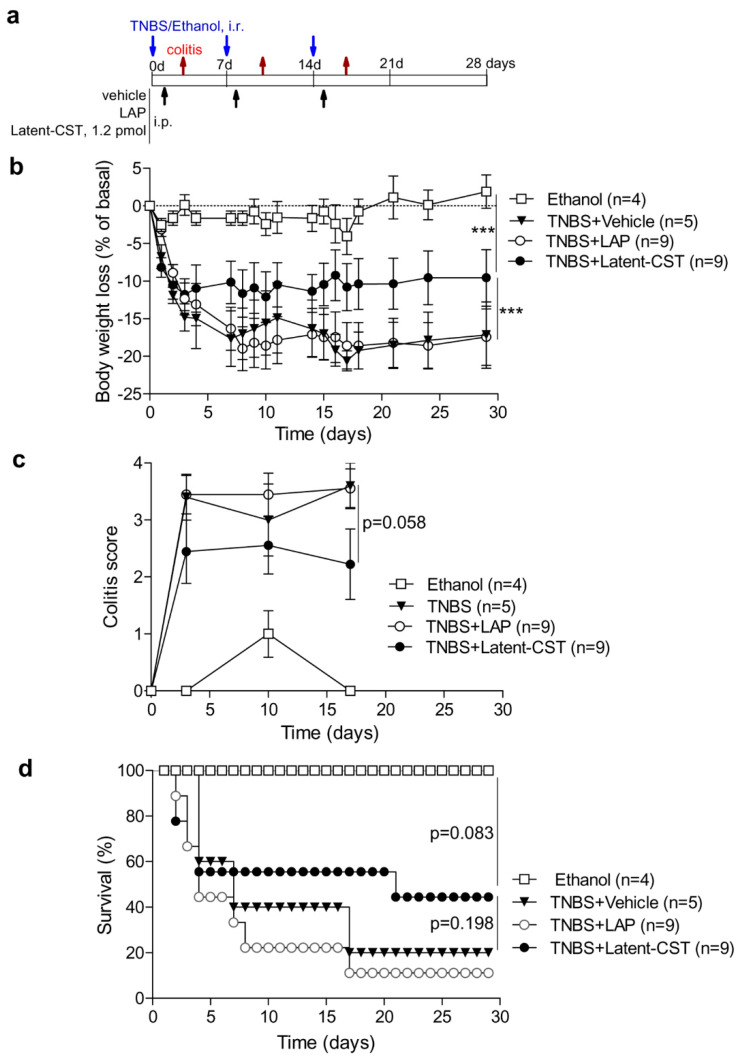
Treatment with Latent-CST protected from chronic inflammatory bowel disease (IBD). (**a**) Chronic colitis was induced by repetitive intrarectal (i.r.) administrations of trinitrobencene sulfonic acid (TNBS) in 50% ethanol and treated with LAP or Latent-CST as described in the scheme. Animals injected with 50% ethanol were used as basal controls of reference. Clinical evolution and disease severity was monitored by measuring body weight changes (**b**), colitis score (**c**), and survival rate (**d**) at the indicated time points. n = 4–9 mice/group. Repeated measures ANOVA test (*p* < 0.001) and post-hoc Bonferroni’s test (*** *p* < 0.001) in panel **b**. Non-parametric repeated measures ANOVA-Friedman test (*p* < 0.05) between all groups in panel **c** (*p*-value in the figure corresponds to the comparison between LAP and Latent-CST groups using Wilcoxon matched-pair signed-rank test). Kaplan–Meier log-rank test in panel **d**. Results show the mean ± SEM.

**Figure 5 pharmaceutics-14-02785-f005:**
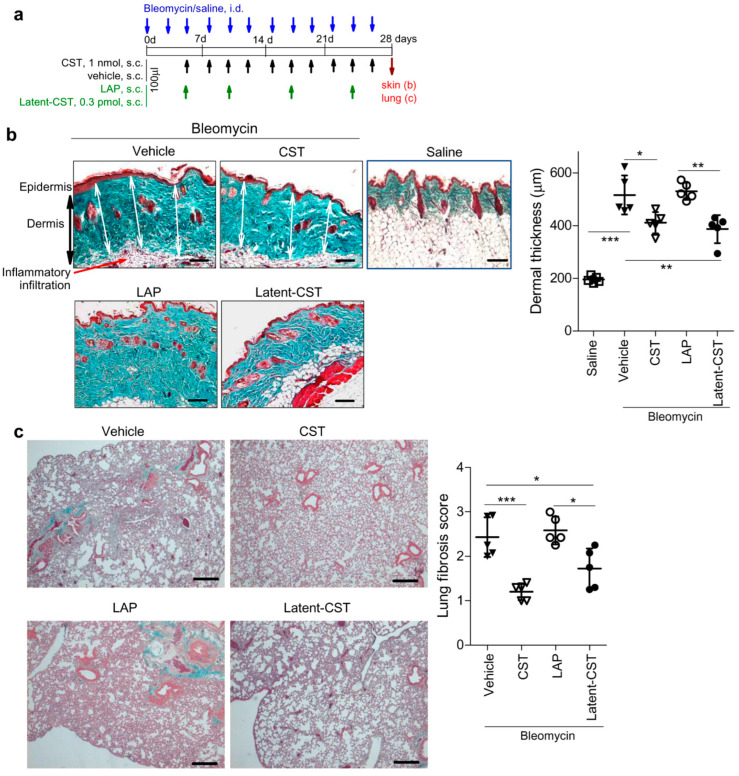
Latent cortistatin shows the same therapeutic efficacy as naïve cortistatin in an experimental model of scleroderma. (**a**) Skin fibrosis was induced in mice by repetitive intradermal (i.d.) injections of bleomycin and treated with vehicle or with naïve cortistatin (CST) or Latent cortistatin (Latent-CST) as described in the scheme. The skin biopsies and lung lobules were collected at day 28. (**b**) Histopathological signs of dermal fibrosis were determined in Masson’s trichrome-stained skin sections (representative images are shown; scale bar: 150 µm). The severity of fibrotic lesions was determined by measuring dermal thickness, which is the distance in µm from the base membrane of epidermal layer to the hypodermal junction with subcutaneous fat, determined as the mean of three random measurements in each section (two examples are showed as white bidirectional arrows) and two sections per mouse. Collagen deposition is evidenced by green staining in sections. Hypodermal inflammatory infiltration is pointed out by a red arrow. See Appendix A for histopathological details using vehicle and Latent-CST-treated mice as examples. Saline-injected skin biopsies were used as basal controls. n = 5 mice/group, one-way ANOVA (*p* < 0.001) and post-hoc Bonferroni’s test, * *p* < 0.05, ** *p* < 0.01 and *** *p* < 0.001. (**c**) Histopathological signs of pulmonary fibrosis were scored in Masson’s trichrome-stained lung sections (n = 5 mice/group, representative images are shown; scale bar: 200 µm). See Appendix A for histopathological details using vehicle and Latent-CST-treated mice as examples. One-way ANOVA Kruskal–Wallis test (*p* < 0.01) and post-hoc Dunn’s multiple comparisons, * *p* < 0.05 and *** *p* < 0.001. Results show the mean ± SD with dots representing individual values of biologically independent animals.

**Figure 6 pharmaceutics-14-02785-f006:**
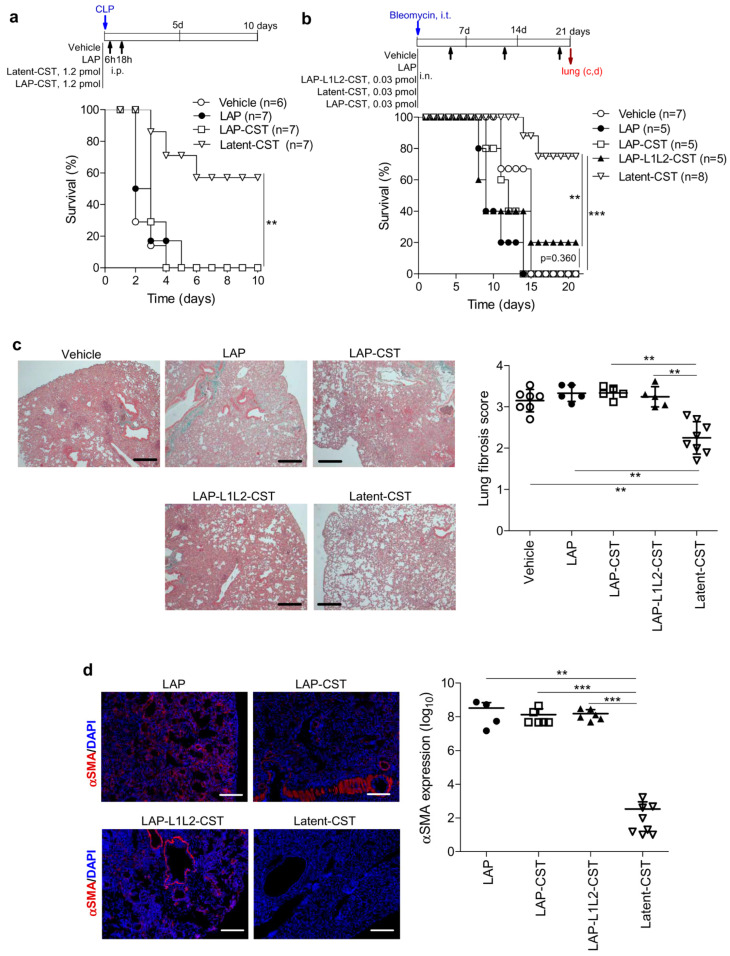
The MMP-cleavage site is essential for Latent-CST to ameliorate inflammation and fibrosis. (**a**) Severe sepsis was induced in mice by CLP and treated with vehicle, LAP, Latent-CST, or LAP-CST as indicated in the scheme, and survival was monitored daily (n = 6–7 mice per group; ** *p* < 0.01 versus LAP-treated mice using Kaplan–Meier log-rank test). (**b**–**d**) Severe pulmonary fibrosis was induced in mice by intratracheal (i.t.) injection of bleomycin and then i.n. treated with vehicle, LAP, LAP-CST, LAP-L1L2-CST, or Latent-CST as indicated in the scheme. (**b**) Mortality (5–8 mice per group; ** *p* < 0.01 and *** *p* < 0.001 versus LAP-treated mice using Kaplan–Meier log-rank test) was monitored daily. (**c**) Pulmonary fibrosis was scored by quantifying the histopathological signs in Masson’s trichrome-stained sections (scale bars: 200 µm) of lungs collected at day 21 (Latent-CST-treated group) or immediately after mice die (all experimental groups). See Appendix A for histopathological details using vehicle and Latent-CST-treated mice as examples. n = 5–8 mice per group, one-way ANOVA Kruskal–Wallis test (*p* < 0.01) and post-hoc Dunn’s multiple comparisons, ** *p* < 0.01. (**d**) The immunofluorescence analysis of α-smooth muscle actin (αSMA)-positive (red) myofibroblasts in lung sections (scale bars: 100 µm) collected at day 21 (Latent-CST-treated group) or immediately after mice die (all experimental groups) was used for quantifying fibrosis markers (data show fluorescence units expressed as log10). n = 5–8 mice per group, one-way ANOVA (*p* < 0.001) and post-hoc Bonferroni test, ** *p* < 0.01 and *** *p* < 0.001. Results show the mean ± SD with dots representing individual values of biologically independent animals.

**Table 1 pharmaceutics-14-02785-t001:** Sequence for double-stranded DNAs that were used to generate the pLAP-CST and pLatent-CST plasmids from the pLAP plasmid.

pLAP-CST	
Forward Primer	5′TCTGCAAAGC**GAATTC**CAGGAAAGACCACCCCTCCAGCAGCCCCCACACCGGGATAAAAAGCCCTGCAAGAACTTCTTCTGGAAAACCTTCTCCTCGTGCAAGTAG**GAATTC**TGCAGATATC3′
Reverse Primer	5′GATATCTGCA**GAATTC**CTACTTGCACGAGGAGAAGGTTTTCCAGAAGAAGTTCTTGCAGGGCTTTTTATCCCGGTGTGGGGGCTGCTGGAGGGGTGGTCTTTCCTG**GAATTC**GCTTTGCAGA3′
pLAP-L1L2-CST	
Forward Primer	5′TCTGCAAAGC**GAATTC**GGGGGAGGCGGATCCGGGGGAGGGGGCTCAGCGGCCGCCCAGGAAAGACCACCCCTCCAGCAGCCCCCACACCGGGATAAAAAGCCCTGCAAGAACTTCTTCTGGAAAACCTTCTCCTCGTGCAAGTAG**GAATTC**TGCAGATATC3′
Reverse Primer	5′GATATCTGCA**GAATTC**CTACTTGCACGAGGAGAAGGTTTTCCAGAAGAAGTTCTTGCAGGGCTTTTTATCCCGGTGTGGGGGCTGCTGGAGGGGTGGTCTTTCCTGGGCGGCCGCTGAGCCCCCTCCCCCGGATCCGCCTCCCCC**GAATTC**GCTTTGCAGA3′
pLatent-CST	
Forward Primer	5′TCTGCAAAGC**GAATTC**GGGGGAGGCGGATCCCCGCTCGGGCTTTGGGCGGGGGGAGGGGGCTCAGCGGCCGCCCAGGAAAGACCACCCCTCCAGCAGCCCCCACACCGGGATAAAAAGCCCTGCAAGAACTTCTTCTGGAAAACCTTCTCCTCGTGCAAGTAG**GAATTC**TGCAGATATC3′
Reverse Primer	5′GATATCTGCA**GAATTC**CTACTTGCACGAGGAGAAGGTTTTCCAGAAGAAGTTCTTGCAGGGCTTTTTATCCCGGTGTGGGGGCTGCTGGAGGGGTGGTCTTTCCTGGGCGGCCGCTGAGCCCCCTCCCCCCGCCCAAAGCCCGAGCGGGGATCCGCCTCCCCC**GAATTC**GCTTTGCAGA3′

The double-stranded (forward and reverse) sequences cloned in pLAP-CST, pLAP-L1L2-CST, and pLatent-CST plasmids are indicated. Grey: sequence for recombination and the *Eco*RI restriction enzyme cleavage site (bold-underlined). Yellow: sequence for L1 and L2 flexible linkers. Red: sequence for cleavage site recognized by metalloproteinases (MMPs). Green: sequence for cortistatin-29.

## Data Availability

The data that support the findings of this study are available from the corresponding author upon reasonable request.

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
