# Peer review of "Therapeutic Effect of a Latent Form of Cortistatin in Experimental Inflammatory and Fibrotic Disorders"

_pharmaceutics, 2022, doi:10.3390/pharmaceutics14122785_

Round 1

Reviewer 1 Report

In this manuscript, the authors synthesized a new cortistatin-based prodrug through a molecular engineering method. This prodrug includes bioactive cortistatin, a molecular-shield provided by the latency-associated protein of the transforming growth factor-β1 and a cleavage site specifically recognized by metalloproteinases that are abundant in inflammatory/fibrotic foci. Moreover, the authors demonstrated the protective effect of this latent form of corticostatin in different disease models. This manuscript proposed an innovative method to improve the stability, bioavailability, and target specificity of corticostatin, which promised to promote the highly efficient therapeutic approaches of bioactive peptides. I recommend publishing after addressing the following major concerns.

1. The authors described in the method “Culture supernatants were collected 48h after cell transfection, assayed for the contents of cortistatin-29 by using a specific competitive ELISA for cortistatin”. However, there is no relevant data in the manuscript indicating the expression of cortistatin. Please authors add the expression of cortistatin mediated by different prodrug.

2. In the chronic colitis and skin fibrosis model, the authors compared the therapeutic effect of Latent-CST with the LAP group (without corticostatin and MMP-cleavage site) and suggested that Latent-CST and LAP-CST (MMP-cleavage site) be compared to demonstrate the importance of response release corticostatin.

3. Both Figure 2 and Figure 5a investigate the therapeutic effects of different prodrugs in sepsis, and Figure 5a investigates the importance of the MMP-cleavage site, which the authors recommend to combine for ease of reading and understanding.

4. The data did not well support the conclusion. For example, the authors designed a latent form of corticostatin in order to improve the half-life and stability of corticostatin; however, the blood half-life and stability of prodrug-mediated corticostatins were not studied in this article.

5. The authors should perform statistical analysis in Figure 3c and Figure 3d.

Author Response

Reviewer 1:

Comments and Suggestions for Authors:
In this manuscript, the authors synthesized a new cortistatin-based
prodrug through a molecular engineering method. This prodrug includes
bioactive cortistatin, a molecular-shield provided by the
latency-associated protein of the transforming growth factor-β1 and acleavage site specifically recognized by metalloproteinases that are
abundant in inflammatory/fibrotic foci. Moreover, the authors
demonstrated the protective effect of this latent form of
corticostatin in different disease models. This manuscript proposed an
innovative method to improve the stability, bioavailability, and
target specificity of corticostatin, which promised to promote the
highly efficient therapeutic approaches of bioactive peptides. I
recommend publishing after addressing the following major concerns.

-Concern 1. The authors described in the method “Culture supernatants were
collected 48h after cell transfection, assayed for the contents of
cortistatin-29 by using a specific competitive ELISA for cortistatin”.
However, there is no relevant data in the manuscript indicating the
expression of cortistatin. Please authors add the expression of
cortistatin mediated by different prodrug.

Response: We have followed the recommendation of this reviewer and we have included a new Figure 2 that describes the contents of cortistatin in the different latent forms in culture supernatants of HEK293 cells transfected with pLAP, pLAP-CST, pLAP-L1L2-CST and pLatent-CST after 24h and 48h culture (Figure 1a). The corresponding methodology is described in Methods section (page 6).  

-Concern 2. In the chronic colitis and skin fibrosis model, the authors
compared the therapeutic effect of Latent-CST with the LAP group
(without corticostatin and MMP-cleavage site) and suggested that
Latent-CST and LAP-CST (MMP-cleavage site) be compared to demonstrate
the importance of response release corticostatin.

Response: We appreciate this suggestion, but unfortunately we were not able to perform this additional experiment. The Animal Care and Use Board and the Ethical Committee of Spanish Council of Scientific Research did not approve the extension of this study to perform this new experiment, arguing that in base of 3R rules (especially reduction and refinement), of the severity of these models and considering that these controls without MMP-cleavage site were successfully used in two of the proposed experimental models, there is no justification to increase the suffering of more animals. We agree with this decision, and in any case, we consider that the demonstration of absence of significant effect of MMP-deleted latent cortistatin in two different models of chronic and severe inflammation and fibrosis is solid. In addition, we have included new data with an additional control (lacking MMP-cleavage site, while keep linkers L1L2, named LAP-L1L2-CST) in experimental lung fibrosis from experiments that were previously performed.

-Concern 3. Both Figure 2 and Figure 5a investigate the therapeutic effects of
different prodrugs in sepsis, and Figure 5a investigates the
importance of the MMP-cleavage site, which the authors recommend to
combine for ease of reading and understanding.

Response: We appreciate this suggestion and found that could be initially logic, but with the addition of new data that try to reinforce the involvement of MMP-cleavage site in the therapeutic activity of the latent form of cortistatin, and the new data that evaluate the stability and cleavage of this formulation in vitro, we consider that it is necessary joint all these data in an specific subsection focused on the potential involvement of MMP-cleavage.

-Concern 4. The data did not well support the conclusion. For example, the
authors designed a latent form of corticostatin in order to improve
the half-life and stability of corticostatin; however, the blood
half-life and stability of prodrug-mediated corticostatins were not
studied in this article.

Response: We have included in a new Figure 2b showing the stability of cortistatin in the different latent forms as a time-course in the present of serum-supplemented medium. We also identify as a limitation of our study that the half-life and tissue distribution of the latent form of cortistatin after its administration in vivo was not investigated, being an important subject that needs to be further addressed as part of the preclinical characterization of this new formulation.  

-Concern 5. The authors should perform statistical analysis in Figure 3c and Figure 3d.

Response: We have followed the recommendation of this reviewed and the statistic analysis for former Figures 3c and 3d (new Figures 4c and 4d) was included.

Reviewer 2 Report

This manuscript describes the significant therapeutic benefits of newly formulated cortistatin (CST) for the animal models of inflammatory and fibrotic disorders. The new formulation includes prodrug design with a molecular shield and MMP proteinase cleavage site. These enable the active peptide cortistatin to be secreted from the prodrug at MMP-abundant sites with inflammation and fibrosis. The manuscript found much less effective dose of this ‘Latent-cortistatin’ compared to the effective dose of naïve cortistatin in animal models. The reviewer found major concerns as follows.

11.   The manuscript used cell culture supernatant that includes cortistatin (CST) for animal studies. Using cell culture supernatant is worrisome to make a conclusion of CST-mediated in vivo benefits, although the manuscript used appropriate control (LAP-CST). Other growth factors or cytokines can be released from the cells and using supernatant can confuse the interpretation of in vivo results. Additional information or previous reports should be added to justify using the supernatant.

22.       The reviewer requests the detail of how CST concentration was measured from cell culture supernatant with ELISA and the results with standard curves should be provided as at least supplemental data. The concentration of cortistatin in supernatant was measured ~ 10ng/mL with ELISA. However, based on Figure 1a, Latent-CST or LAP-CST appears to be somewhat buried with LAP (latency-associated peptide). This was also consistent with Figure 5 where LAP-CST showed no effects for animal models. The ELISA may detect CST level incorrectly. If the CST concentration from the supernatant is inaccurately measured, this will markedly diminish the benefit of using Latent-CST (for example, much low dose of Latent-CST was used to achieve the similar benefit in animal models). Interestingly, if ELISA can detect CST from LAP-CST, LAP-CST should be exposed somewhat to the specific antibody. This suggests that LAP-CST may be available for the interaction with receptor molecules (this contrasts with Figure 5). In addition, how cells were cultured and were transfected with DNA plasmids and the culture conditions (including FBS) should be added to Method section.

33.       Data representation in this manuscript should be improved.

a.       Indicate how big LAP (molecular weight or size) is in Figure 1a.

b.       For figure 2, provide explanation of how lung biopsy (dissection) and serum collection from cardia puncture themselves at 48h will affect the animal’s survival rates. The reviewer quite doesn’t understand how these mice can live properly with these significant damage to lung and heart.   

c.       Indicate animal number in each of the survival graphs. This will help understand better.

d.       Use mean and S.D (not SEM) in figure representation to show actual variation in the group. SEM will skew the variation since increasing n number will reduce the error bars. Also show all data points of each animal with mean and S.D (at least all bar graphs). This representation becomes more common for animal studies to clearly show variation in the groups and how many animals are used for figure representation.

e.       Statistical analysis should be improved. Use ANOVA for multi-group comparison for example, figure 2C bar graph. Justify the use of non-parametric analysis versus parametric.  Most concentration values from blood sample will belong to parametric analysis (for example, Figure 2d).

f.        Mark the difference in representative histological pictures with an arrow in all figures (in particularly, figure 4b and 4c). For some figures, using arrow will markedly increase reader’s understanding.

g.       Figure 4b, please show a histological picture of the saline control used for dotted line in the bar graph. Please indicate the dermal region in the histological pictures. This will increase the clarity.  

h.       Figure 5b, please indicate S.D. of mean score under histological images.

i.         Quantify aSMA expression level in Figure 5b.

44.       The effectiveness of Latent-CST and MMP involvement needs more data and explanation. Just showing results of the LAP-CST group in Figure 5 is insufficient to argue that MMP-mediated CST release from the prodrug is essential for in vivo benefits. The reviewer requests additional data such as results from a relatively simple in vitro assay (such as cell-based system to analyze receptor activation by CST) where the authors can show that CST is released from Latent-CST with addition of MMP or the release is inhibited by MMP inhibitors. These results will strengthen the benefit of using Latent-CST.

55.       The conclusion section needs more discussion points regarding  

a.       Potential toxicity of LAP only group should be addressed and discussed. Please see Figure 2b and 3d.

b.       Potential receptors working for CST and mechanism of action

c.       Limitations of this study and future direction.

66.       Other points

a.       ‘Similar… than’ should be fixed with ‘similar… to’. This quite confused the reviewer.

b.       More appropriate controls for Latent-CST would be LAP-linker1-null MMP cleavage site(by mutations or 6 random amino acids)-linker2-CST.

c.       Line 322, this sentence is a bit odd because there is no data about lethargy, diarrhea, body weight loss and hypothermia. It appears that this sentence was carried over from some other manuscripts.

d.       Line 331, please provide results that support the efficacy at 6h after sepsis induction. Figure 2 does not have any data at 6h showing any benefit of Latent-SCT.

e.       Line 346, it shows four cycles of intracolonic infusion. But figure 3a only shows three arrows for i.r. infusion.

f.        Line 537, intranasal injection should be fixed since no intranasal injection for Figure 4.

g.       Line 613, please provide known Kd values of CST for its receptors.

h.       Line 616, please fix ‘proven’ with ‘shown’.

Author Response

Reviewer 2

Comments and Suggestions for Authors
This manuscript describes the significant therapeutic benefits of
newly formulated cortistatin (CST) for the animal models of
inflammatory and fibrotic disorders. The new formulation includes
prodrug design with a molecular shield and MMP proteinase cleavage
site. These enable the active peptide cortistatin to be secreted from
the prodrug at MMP-abundant sites with inflammation and fibrosis. The
manuscript found much less effective dose of this ‘Latent-cortistatin’
compared to the effective dose of naïve cortistatin in animal models.The reviewer found major concerns as follows.

-Concern 11.   The manuscript used cell culture supernatant that includes
cortistatin (CST) for animal studies. Using cell culture supernatant
is worrisome to make a conclusion of CST-mediated in vivo benefits,
although the manuscript used appropriate control (LAP-CST). Other
growth factors or cytokines can be released from the cells and using
supernatant can confuse the interpretation of in vivo results.
Additional information or previous reports should be added to justify
using the supernatant.

Response: We appreciate this comment and fully agree with this reviewer that this issue should be discussed in the manuscript. We have partially justified the use of HEK293 cell supernatants in the Methods section (page 6) and discussed in comparison with other studies in the end of the first paragraph of the discussion section (pages 16-17). Moreover, the fact that from all the latent forms of cortistatin that we used in this study, only Latent-CST (that is the unique that is able to potentially release cortistatin in inflammatory and fibrotic foci), but not LAP-CST and LAP-L1L2-CST, was able to show significant in vivo effects, also supports that other soluble factors secreted by HEK293 cells were not significantly contributing to the therapeutic effect of these culture supernatants.

-Concern 22. The reviewer requests the detail of how CST concentration
was measured from cell culture supernatant with ELISA and the results
with standard curves should be provided as at least supplemental data.
The concentration of cortistatin in supernatant was measured ~ 10ng/mL
with ELISA. However, based on Figure 1a, Latent-CST or LAP-CST appears
to be somewhat buried with LAP (latency-associated peptide). This was
also consistent with Figure 5 where LAP-CST showed no effects for
animal models. The ELISA may detect CST level incorrectly. If the CST
concentration from the supernatant is inaccurately measured, this will
markedly diminish the benefit of using Latent-CST (for example, much
low dose of Latent-CST was used to achieve the similar benefit in
animal models). Interestingly, if ELISA can detect CST from LAP-CST,
LAP-CST should be exposed somewhat to the specific antibody. This
suggests that LAP-CST may be available for the interaction with
receptor molecules (this contrasts with Figure 5). In addition, how
cells were cultured and were transfected with DNA plasmids and the
culture conditions (including FBS) should be added to Method section.

Response: We have included in the method section how cortistatin was determined in culture supernatants of HEK293 cells transfected with the different plasmids, including the conditions of transfection and culture (page 6). These results are included in a new figure 2 describing the content of cortistatin in the different latent forms (using LAP as negative control), as well as the stable presence of cortistatin in these forms in the presence of serum. We agree with this reviewer that a discussion about how ELISA is able to detect the presence of cortistatin in the latent form while is shielded by LAP, and at the same time avoid its degradation by peptidases and its interaction with its receptors. These issues are discussed in the context of other studies that used the same strategy in the first paragraph of the discussion section (page 16).

-Concern 33.       Data representation in this manuscript should be improved.

a.       Indicate how big LAP (molecular weight or size) is in Figure 1a.

Response: The number of aminoacids of LAP has been included in figure 1a, in order to take it as reference since the scheme including all the functional elements of the molecule (linkers, MMP-cleavage site, and cortistatin) is not in scale with LAP.

b.       For figure 2, provide explanation of how lung biopsy
(dissection) and serum collection from cardia puncture themselves at
48h will affect the animal’s survival rates. The reviewer quite
doesn’t understand how these mice can live properly with these significant damage to lung and heart.

Response: We are sorry with the writing of this part of the methods that could induce to confusion to this reviewer. Lungs, serum and peritoneal suspension were collected from a set of animals different to those used for survival. This means that for each experimental group we used a set of animals for monitoring mortality and clinical signs, and a separate set of animals for collecting serum, lungs and peritoneal suspension after their sacrifice. We have rewritten this sentence in the method section in order to clarify this subject.

c.       Indicate animal number in each of the survival graphs. This
will help understand better.

Response: We have followed this suggestion, and number of animals has been included in all survival graphs.

d.       Use mean and S.D (not SEM) in figure representation to show
actual variation in the group. SEM will skew the variation since
increasing n number will reduce the error bars. Also show all data
points of each animal with mean and S.D (at least all bar graphs).
This representation becomes more common for animal studies to clearly
show variation in the groups and how many animals are used for figure
representation.

Response: We have followed all these suggestions. All graphs of the manuscript now show the mean and SD, with the exception of time-curves of IBD model (Figure 4b and 4c), that overlapping in the error bars between the multiple experimental groups made difficult the visualization of the data, and for clarity we used mean and SEM. Moreover, we have substituted the data on columns by points corresponding to individual animals or experiments (in vitro) in all graphs, with the exception of survival or complex time-curves (Figure 4b and 4c).

e.       Statistical analysis should be improved. Use ANOVA for
multi-group comparison for example, figure 2C bar graph. Justify the
use of non-parametric analysis versus parametric.  Most concentration
values from blood sample will belong to parametric analysis (for
example, Figure 2d).

Response: Following the suggestions of this reviewer, we have included more details about the statistic analysis in the method section and figure legends, including the justification of parametric versus non-parametric analysis. ANOVA was applied for multi-group comparisons.

f.        Mark the difference in representative histological pictures
with an arrow in all figures (in particularly, figure 4b and 4c). For
some figures, using arrow will markedly increase reader’s understanding.

Response: We fully agree with this suggestion. We have included relevant information in main figures, when possible, and we have generated three new supplementary figures that include enlarged figures of skin and lung sections, as examples, highlighting the most relevant pathological signs.

g.       Figure 4b, please show a histological picture of the saline
control used for dotted line in the bar graph. Please indicate the
dermal region in the histological pictures. This will increase the
clarity.

Response: These suggestions have been included in the new Figure 5b (former figure 4b)

h.       Figure 5b, please indicate S.D. of mean score under
histological images.

Response: We have followed this suggestion, with a new figure 6c (former figure 5b) including fibrosis scores of individual animals, with mean and SD and statistical analysis between groups.

i.         Quantify aSMA expression level in Figure 5b.

Response: We have followed this suggestion, with a new figure 6d (former figure 5b) including quantification of aSMA expression of individual animals and statistic analysis.

-Concern 44.       The effectiveness of Latent-CST and MMP involvement needs
more data and explanation. Just showing results of the LAP-CST group
in Figure 5 is insufficient to argue that MMP-mediated CST release
from the prodrug is essential for in vivo benefits. The reviewer
requests additional data such as results from a relatively simple in
vitro assay (such as cell-based system to analyze receptor activation
by CST) where the authors can show that CST is released from
Latent-CST with addition of MMP or the release is inhibited by MMP
inhibitors. These results will strengthen the benefit of using
Latent-CST.

Response: We agree with this reviewer that it is convenient a further discussion about the potential involvement of MMP-site in the activity of the latent form of cortistatin. We have tried to address this issue in the discussion section (page 17, second paragraph). The experiment that is proposed by the reviewer is a logic and good proof of concept to demonstrate in vitro the bioactivity of the released cortistatin by MMP. Unfortunately, we have not experience with this methodology and setting this experiment in our laboratory would take a long period of time, well beyond the time given by the editor to submit a revised version of the manuscript. Alternatively, we have included an additional experiment in which we indirectly evaluated in vitro the release of cortistatin from the latent form in the presence of MMP1 (Methods section: page 6, last paragraph; Result section: page 14, last paragraph). In any case, we have identified as a limitation of our study the fact that we have not demonstrated in vitro that MMP-induced released cortistatin is able to signal through specific receptors (see Discussion section, page 17).

-Concern 55. The conclusion section needs more discussion points regarding

a.       Potential toxicity of LAP only group should be addressed and
discussed. Please see Figure 2b and 3d.

Response: The statistical analysis indicates that LAP did not show significant differences with vehicle-treated group in all the experimental models. Therefore, in our opinion there are no reasons to believe that treatment with LAP results in a toxic effect in the animal.  

b.       Potential receptors working for CST and mechanism of action.

Response: We have followed this suggestion and we have included discussion about the potential receptors and signal transduction involved in the effect of latent cortistatin in immune and fibrotic cells (page 17).

c.       Limitations of this study and future direction.

Response: We have followed this suggestion and a number of limitations of our study have been included in the discussion section as well as important aspects that need to be addressed in further preclinical characterization of this new formulation before initiating clinical studies in animal or human immune/fibrotic disorders.

-Concern 66.       Other points

a.       ‘Similar… than’ should be fixed with ‘similar… to’. This
quite confused the reviewer.

Response: This recurrent edition mistake has been corrected in the new version of the manuscript. We appreciate that this reviewer identifies it.

b.       More appropriate controls for Latent-CST would be
LAP-linker1-null MMP cleavage site (by mutations or 6 random amino
acids)-linker2-CST.

Response: We appreciate this suggestion, and we consider that is a good additional control. However, the design, synthesis and transfection of this new control and its assay in at least a chronic model of inflammation or fibrosis will take a long time, well beyond the time given by the editor to submit a revised version of the manuscript. Moreover, the Animal Care and Use Board and the Ethical Committee of Spanish Council of Scientific Research did not approve the extension of the study with more animals for assaying an additional control, arguing in base of 3R consensus, that there is no justification to increase the suffering of more animals in severe chronic models and considering that controls without MMP-cleavage site were successfully used in two of the proposed experimental models. To reinforce that MMP-cleavage site is essential for latent cortistatin to exert its therapeutic actions, we have included an additional latent form as control that was compared in the previous study design, consisting in LAP-linkersL1L2-CST, while lacking MMP-site. These data have been included in Figure 6b-d. In any case, we have tried to tone down the conclusion that cortistatin needs to be released form the latent form for exert its actions in vivo, instead focusing in the fact that the presence of MMP-site in the molecule is critical for is therapeutic effects.

c.       Line 322, this sentence is a bit odd because there is no data
about lethargy, diarrhea, body weight loss and hypothermia. It appears
that this sentence was carried over from some other manuscripts.

Response: We have included a new figure (Figure 3d) including the data of scores for these different clinical signs.

d.       Line 331, please provide results that support the efficacy at
6h after sepsis induction. Figure 2 does not have any data at 6h
showing any benefit of Latent-CST.

Response: Probably there is some confusion with this sentence. We are trying to highlight that the treatment with latent CST started 6h after CLP induction (see scheme in figure 3a). All samples were collected 48h after CLP, but not at 6h.

e.       Line 346, it shows four cycles of intracolonic infusion. But
figure 3a only shows three arrows for i.r. infusion.

Response: We are sorry for this mistake. The correct sentence is three cycles of TNBS infusion.

f.        Line 537, intranasal injection should be fixed since no
intranasal injection for Figure 4.

Response: Again we apologize for this mistake, and appreciate that this reviewer identifies it.

g.       Line 613, please provide known Kd values of CST for its receptors.

Response: The range of Kd values for receptors involved in cortistatin effects in immune and fibrotic cells have been included in the discussion section (page 17)

h.       Line 616, please fix ‘proven’ with ‘shown’.

Response: “Proven” has been fixed with “shown”.

Reviewer 3 Report

Campos-Salinas et al., prepared well organized and well written manuscript. the quality of presentation and scientific soundness is satisfying.. some minor comments should be addressed by authors before consideration for publication.

- Please update references prior to 2010 as possible

- Line 133 "Mice were allowed to acclimatize to the experimental 133 room for 1 h before experiments" did you think 1 hour is enough for acclimization? can you add reference? 

- Line 134 "Mice were randomly assigned to the different experimental groups" ... please mention the total number of mice and their classification and number in each group. Please add groups and their abbreviations in clear manner.

- Line 224 "using standard techniques"... add reference here.

- Line 235... add number of mice used.

- Line 288 "Severe lung fibrosis was induced in...." please add reference for the induction method.

- Figure 5b... please add figure for control (induced fibrosis without treatment) ... i can't identify the color of Masson's staining... can you replace with clear figures?

- Conclusion is too long... please revised and rewrite in brief.

- Please remove the inappropriate self citations.

Author Response

Reviewer 3

Comments and Suggestions for Authors
Campos-Salinas et al., prepared well organized and well written
manuscript. the quality of presentation and scientific soundness is
satisfying.. some minor comments should be addressed by authors before
consideration for publication.

- Please update references prior to 2010 as possible.

Response: The reference list has been updated when possible. Only essential papers that describe methodologies or seminal papers that describe important aspects of the cortistatin functions prior 2010 were included.

- Line 133 "Mice were allowed to acclimatize to the experimental 133
room for 1 h before experiments" did you think 1 hour is enough for
acclimization? can you add reference?

Response: We believe that this sentence aroused some confusion in this reviewer. We have rewritten it in order to clarify it (page 4).

- Line 134 "Mice were randomly assigned to the different experimental
groups" ... please mention the total number of mice and their
classification and number in each group. Please add groups and their
abbreviations in clear manner.

Response: Because the numbers of animals are very diverse in the different experimental groups, depending of the various experimental models, we consider that is extremely difficult to mention in this point of the manuscript the total number of animals and the assignation to each group. Moreover, because the different plasmids that we used to transfect the cells and generate the different latent forms and controls have not been still defined in this point of the manuscript, we also consider that is difficult to add the groups and their abbreviations in this subsection. Alternatively, we have decided to name the groups and abbreviations in the subsection where the latent forms are explained, and the number of animals for each group and experimental model has been included in the figure legends and graphs (when possible, as individual dots). In any case, this issue is mentioned in the subsection of “The Animals and ethic statements”.  

- Line 224 "using standard techniques"... add reference here.

Response: It has been referenced in the new version of the manuscript.

- Line 235... add number of mice used.

Response: As mentioned above, the number of animals for each experimental model was included in the corresponding figure legends and graphs.

- Line 288 "Severe lung fibrosis was induced in...." please add
reference for the induction method.

Response: This method was referenced in the new version of the manuscript.

- Figure 5b... please add figure for control (induced fibrosis without
treatment) ... i can't identify the color of Masson's staining... can
you replace with clear figures?

Response: We have followed this suggestion and the vehicle-treated control was included in a new figure 6b and 6c (former 5b). We have included new pictures of sections with higher resolution and contrast. Moreover, we have scored the histopathological signs and we have included new Supplementary figures with examples (vehicle- and Latent-CST-treated mice) showing images with increased sizes and resolution. In any case, the original figures have higher quality than that the inserted in the pdf version of the manuscript that is sent for review.

- Conclusion is too long... please revised and rewrite in brief.

Response: Because the other reviewers suggested additional discussion of the manuscript, we have reedited the manuscript, leaving the results section alone (we previously combine result and discussion in a single section), adding a separate section for discussion, and deleting the conclusion section that we agree with this reviewer that it was too long.

- Please remove the inappropriate self citations.

Response: We have removed various of our references, leaving only those are essential for understanding the manuscript.

Round 2

Reviewer 1 Report

I would like to thank the authors for addressing the majority of my comments. The manuscript is now easier to read and understand, and the quality of the figures has improved.

Author Response

We really appreciate the positive and constructive words of this reviewer and that he/she has found significant improvements in the new version of the manuscript.

Reviewer 2 Report

Reviewer’s comment on the revised manuscript.

The revised manuscript has been markedly improved addressing most of the concerns raised by the reviewer. However, the statistical analysis still needs lots of attention and there is a significant discrepancy between the method section and figure legends.

The authors mentioned they used ANOVA for multiple comparisons either with parametric (continuous variable) or non-parametric (histological scores) variables in the method section. But in the figure legions of Figure 3c, 3d, Man-Whitney U-test was used that is for two group comparison. In figure 4b and 4c, Wilcoxon-matched-pair signed rack test was used that is inappropriate for figure 4b and 4c. The reviewer believes that repeated measure ANOVA for figure 4b and non-parametric repeated measure ANOVA (Friedman Test) for figure 4c are appropriate statistical analyses. In figure 5b and 5c, the authors used student t-test and Mann-Whiteny U-test that are for two group comparison. ANOVA and Kruskal-Wallis analysis should be used as the authors mentioned in the Method section. In figure 6c and 6d, the authors still used Mann-Whiteny U-test that is for two group comparison. ANOVA should be used.

In my second reading, another important issue came up about comparing doses of cortistatin vs a latent form of cortistatin. Since the authors used only one dose of cortistatin, comparing potency between cortistatin and a latent cortistatin in vivo experiment would not make sense. For figure 3, CST 0.5 nmol was used, but lower doses of CST might also be able to show similar efficacy in vivo model. Although 400-fold lower dose of a latent form of CST (1.2pmol) showed a similar efficacy to 0.5nmol CST, this does not mean a latent form of CST is 400 potent than CST because only one effective dose is used. The authors should be more cautious for mentioning more potency of a latent form of CST compared to CST in animal models.

Minor concern

1. Please indicate how to measure aSMA expression levels such as software, designated area used for measurement, which color was used for measurement, and so on.

2. line 389, remove the sentence that ‘therapeutically effective even starting six hours’. There are no results to support this. Just mention the treatment was introduced at 6h after sepsis induction.  

3. Figure 3 legend c and e, remove ***p<0.001 that is not used.

4. Discussion regarding CST measurement by ELISA appears to have week grounds (starting line 568 to 574). The authors mentioned ‘the estimation of cortistatin contents in the latent form could be quite exact’. But the authors should also mention the possible case that concentration is inaccurately measured and what it affects the whole study and further mention future experiments and experimental design to confirm the concentration of CST secreted from HEK293T cells with all serum components.

5. Please double-check the scale of Figure 5b Saline picture again if the scale is correct for 200 micrometer. It appears to be a bit different from other pictures to my perspective. Also indicate where the authors measured for dermal thickness in the histological picture of figure 5b.

Author Response

We really appreciate the positive words of this reviewer with the new version of the manuscript, as well as his/her new suggestions that significantly will improve the quality of the paper. We have accordingly modified the manuscript to incorporate all the recommendations and corrections raised by this reviewer.

First, we have followed all the suggestions concerning statistical analysis of figures 3-6. In the previous version of the manuscript, we performed both multiple comparisons between groups using ANOVA and comparison between two groups, and in order to do not complicate the graphics with excessive statistical data, we decided mainly including comparison between two groups (mostly LAP vs. Latent-CST or vehicle vs. CST). However, we now agree with this reviewer that inclusion of multiple comparisons between groups is more adequate for this study (see changes in Method Sections and corresponding figure legends).

Second, we also agree with this reviewer that we have to tone down the conclusions about comparative potency between latent and naïve cortistatin in the absence of a complete dose response in the same study. This issue has been discussed in page 17.

Regarding the minor concerns raised by this reviewer:

  1. We have included details about how the immunofluorescence analysis of a-SMA was performed in the study (pages 8 to 9).
  2. We have followed the suggestion of this reviewer of removing the sentence about the potential therapeutic importance of administering latent-CST 6h after the induction of CLP.
  3. Mismatches on ***p<0.001 and others in the figure legends have been adequately corrected.
  4. We have followed the suggestion of this reviewer of adding some additional discussion about the necessity to verify the doses of cortistatin in the injected suspension and culture supernatants of HEK293 in base on the potential errors caused by measuring the peptide levels solely with a competitive ELISA and how this could be limiting the conclusions of the study (see end of page 17).
  5. We have checked the scale for figure 5b, and we confirm that the scale is the same for all the skin sections. However, we have detected a mistake in the number associated to the scale in all the skin panels in the figure legends. The correct scale is 150um instead 200um. Moreover, we have included additional information in the figure legend (see also methods section) of how dermal thickness was measured in the different samples, and we have illustrated with white bidirectional arrows in two of the sections (vehicle- and CST-treated mice) as two representative examples of measurements.

Round 3

Reviewer 2 Report

The reviewer thanks the authors for addressing all concerns mentioned. 

Two minor typos: 

1. line 135, established.

2. line 432, p=0.019, this needs somewhat more information. Is this for whole ANOVA test or a p value from post hoc assay? Also figure 3 shows p=0.058 that is from post- hoc assay. More clarification in the figure legend would help.

Author Response

Thank you again for the positive words of this reviewer and his/her exaustive review that has increased the quality of our manuscript.

1. We have corrected "established" in the new version of the manuscript.

2. We agree with this reviewer that the statistic information for legend of figure 4 could be confusing, specially those related to panel 4c (time curves for colitis score). We have tried to clarify it as follow:

"... Repeated measures ANOVA test (p<0.001) and post-hoc Bonferroni’s test (***p<0.001) in panel b. Non-parametric repeated measures ANOVA-Friedman test (p<0.05) between all groups in panel c (p-value in figure corresponds to the comparison between LAP and Latent-CST groups using Wilcoxon matched-pair signed-rank test). Kaplan-Meier log-rank test in panel d (***p<0.001). Results show the mean±SEM."

Here, we have changed p-value of whole ANOVA p=0.019 for p<0.05 to uniform the format with the other data. Beside the ANOVA analysis of multiple groups, in the figure 3c, we included the exact p-value (p=0.058) for the comparative analysis between only between two groups (LAP vs Latent-CST) using non-parametric Wilcoxon test, in order to show that it was close to reach statistical significance despite being p>0.05.